# Rethinking Resolution in the Context of Efficient Video Recognition

**Chuofan Ma**[*]
The University of Hong Kong
b20mcf@connect.hku.hk

**Qiushan Guo**
The University of Hong Kong
qsguo@cs.hku.hk

**Yi Jiang**[†]
ByteDance Inc.
jiangyi0425@gmail.com

**Ping Luo**
The University of Hong Kong
pluo@cs.hku.hk

**Zehuan Yuan**
ByteDance Inc.
yuanzehuan@bytedance.com

**Xiaojuan Qi**[†]
The University of Hong Kong
xjqi@eee.hku.hk

## Abstract

In this paper, we empirically study how to make the most of low-resolution frames for efficient video recognition. Existing methods mainly focus on developing compact networks or alleviating temporal redundancy of video inputs to increase efficiency, whereas compressing frame resolution has rarely been considered a promising solution. A major concern is the poor recognition accuracy on low-resolution frames. We thus start by analyzing the underlying causes of performance degradation on low-resolution frames. Our key finding is that the major cause of degradation is not information loss in the down-sampling process, but rather the mismatch between network architecture and input scale. Motivated by the success of knowledge distillation (KD), we propose to bridge the gap between network and input size via cross-resolution KD (ResKD). Our work shows that ResKD is a simple but effective method to boost recognition accuracy on low-resolution frames. Without bells and whistles, ResKD considerably surpasses all competitive methods in terms of efficiency and accuracy on four large-scale benchmark datasets, i.e., ActivityNet, FCVID, Mini-Kinetics, Something-Something V2. In addition, we extensively demonstrate its effectiveness over state-of-the-art architectures, i.e., 3D-CNNs and Video Transformers, and scalability towards super low-resolution frames. The results suggest ResKD can serve as a general inference acceleration method for state-of-the-art video recognition. Our code will be available at https://github.com/CVMI-Lab/ResKD.

## 1 Introduction

In recent years, video recognition has gained popularity due to the phenomenal growth of social media platforms and the deluge of online videos. With the introduction of various temporal fusion modules [25, 27, 30, 31], 3D-CNNs [41, 4, 9], and video transformers [34, 2, 35, 29], remarkable progress has been made toward increasing recognition accuracy. Nevertheless, the impressive performance comes at the cost of high computational budgets and latency. State-of-the-art models usually require thousands of GFLOPs of computation to recognize a ten-second video [29], which limits their deployment to resource-restricted devices or latency-constrained applications.

To overcome this limitation, extensive studies [42, 8, 23, 32, 46, 28] have been conducted to increase inference speed on videos. A large group of works [50, 24, 28] concentrates on developing efficient

---

[*]This work was performed when Chuofan Ma worked as an intern at ByteDance.

[†]Corresponding authors

36th Conference on Neural Information Processing Systems (NeurIPS 2022).

frame sampling policy networks to select the most salient frames for recognition in an adaptive manner. Essentially, the idea is to alleviate the inherent temporal redundancy of videos by compressing temporal resolution of the input. However, another crucial factor affecting efficiency, namely spatial resolution of the input, is largely under-explored.

Recent work [45] points out that video contents are spatially redundant as well, providing the theoretical feasibility of compressing spatial resolution to achieve higher efficiency. Nevertheless, in practice, the performance of video recognition models usually degrades drastically on low-resolution inputs, providing a poor trade-off between speed and accuracy. For this reason, spatial down-sampling is rarely considered a promising solution to efficient video recognition. Even in AR-Net [32] which also tries to save computation using low-resolution frames, spatial down-sampling is only used when processing less important frames with particular care. Whereas in this paper, we challenge this common belief by showing that low-resolution frames are far from being fully exploited. In particular, our empirical study reveals the following facts:

**(i) Low-resolution frames are not necessarily low-quality frames.** An intuitive explanation for the performance degradation is the poor quality of input, given the fact that considerable information is lost in the down-sampling process, e.g., the high-frequency signals. However, through controlled experiments, we find that information loss only has mild impacts on accuracy. This observation indicates that substantial redundancy exists in high-resolution (e.g., $224 \times 224$) frames, e.g., task-irrelevant information, while low resolutions mitigate such redundancy and potentially provide higher efficiency.

**(ii) Mismatch between network and input scale leads to sub-optimal performance at low resolutions.** Instead of information loss, the experimental results suggest that scale variation is the main cause of performance degradation. In an investigation into how scale variation influences performance, we notice that network architecture plays an important role. Specifically, we find current models initially designed for high-resolution input can be poor learners at low resolutions. Our observations align with the conclusions drawn in a recent study of network design [40], which posit network architecture should coordinate with resolution to make the most of input.

**(iii) Performance gap between resolutions can be effectively closed via cross-resolution knowledge distillation (ResKD).** Driven by simplicity and generality considerations, we ask whether ResKD could serve as a general solution to alleviate network and input mismatch and unlock the potential of low-resolution frames for efficient video recognition. The idea of ResKD is quite straightforward: a teacher taking high-resolution frames as input is leveraged to guide the learning of the student on low-resolution frames. We benchmark ResKD in terms of effectiveness, generality, and scalability on Kinetics-400 and four popular efficient video recognition datasets, i.e., ActivityNet, FCVID, Mini-Kinetics, and Something Something V2. Extensive experiments show that ResKD outperforms all competitive efficient video recognition methods by large margins. Especially, ResKD uses 4x fewer FLOPs than the recently proposed AdaFocus V2 [46], while achieving 1.4% higher top-1 accuracy on Mini-Kinetics. In addition, we demonstrate the effectiveness of ResKD is agnostic to network architectures. For instance, applying ResKD on Video Swin [29] saves over 1000 GFLOPs per video in inference without sacrificing accuracy.

Contributions of this work are summarized as follows:

- We conduct an in-depth analysis of the underlying causes of performance degradation on low resolution videos. Our study reveals the great potential of low-resolution frames in trade-off for efficiency, which is largely overlooked by current literature.

- We provide a simple but very strong baseline, ResKD, to fulfill the potential of low-resolution frames. In particular, we massively boost the performance of ResKD by identifying and overcoming some major flaws in the current designs of ResKD.

- Extensive experiments are conducted to benchmark ResKD in terms of effectiveness, generality, and scalability. ResKD consistently outperforms state-of-the-art methods on four large-scale action recognition datasets.

## 2   Related Work

**Video recognition.**    The ability to relate frames at different temporal locations is of key importance for video understanding. This motivates recent research on equipping deep neural networks with

temporal modeling capability, which has spawned a cascade of 2D-CNNs, 3D-CNNs, and transformers for video recognition. In modeling temporal information with 2D-CNNs, a common practice is to extract frame-wise features and aggregate them along the temporal dimension, with the help of recurrent neural networks [6, 26] or carefully designed temporal fusion modules [25, 27, 30, 31]. On the other hand, 3D-CNNs, such as C3D [41], I3D [4], and X3D [9], directly extract spatial-temporal representations from videos by densely applying 3D convolutions on sampled clips. As for video transformers, based on ViT, several variants [34, 2, 35, 29] are proposed for video recognition by extending attention mechanism to spatial-temporal patches.

Although the aforementioned models demonstrate impressive recognition accuracy, high computational cost and latency (especially for 3D-CNNs and transformers) restrict their applicability in real-life settings. Efficiency thus comes into the picture. Current works mainly tackle the problem through designing compact networks [43, 51, 53, 42, 8, 23] or dynamically allocating computation to the most informative part of a video [50, 24, 32, 46, 39, 28]. But our approach is orthogonal to these methods, in the sense that it is agnostic to network design and robust to video content.

**Reducing video redundancy.** Videos are inherently redundant medium in carrying information, e.g., there exists considerable still scenes and task-irrelevant clips in untrimmed videos. Based on this observation, various methods have been proposed to mitigate temporal redundancy in video input, such as adaptive frame selection [24, 47, 50, 11, 28], early exiting without watching the full video [7, 48, 13], reusing features from previous frames [33], processing less important frames with lower resolution/precision [32, 39]. Besides, video redundancy is not limited to the temporal dimension. Recently, AdaFocus [45] points out significant spatial redundancy exists in the form of uninformative regions in each frame. Our method shares a similar idea as AdaFocus on reducing spatial redundancy but studies the problem from the perspective of resolution. Moreover, compared with popular adaptive methods [50, 11, 32, 39, 33, 13, 45, 28, 46], our method is easier to implement as it does not contain additional modules or require complex training strategies.

**Knowledge distillation.** Knowledge transfer is pioneered by [1, 17] to distill structure knowledge from teacher network to student network. For the action recognition task, depth [12], temporal information [38], and multiple teacher ensembles [47] are explored as structure knowledge to distill. A high-resolution teacher is first explored in [36] to guide the recognition of extreme low-resolution videos. However, cross-resolution knowledge distillation (ResKD) only demonstrates minor improvements in prior works [36, 39] for video recognition. Our study reveals that lacking consideration of spatial information during distillation leads to a sub-optimal performance of ResKD. By revisiting the design, for the first time, we show ResKD alone massively boosts the efficiency of video recognition.

# 3 A Look into Performance Degradation at Low Resolution

Compressing the spatial resolution of videos has rarely been considered a promising solution to increasing the efficiency of video recognition. This is because the performance of current models usually degrades drastically on low-resolution frames, leading to a poor trade-off between efficiency and accuracy. In this section, we investigate the underlying causes of such performance degradation, aiming to find out how to make the most of low-resolution frames for efficient video recognition. We initiate the exploration from two perspectives: quality of low-resolution frames, and mismatch between network structures and input scales.

**Quality of low-resolution frames.** An intuitive explanation for the underperformance is the poor quality of low-resolution frames. Compared with original high-resolution frames, considerable information, which might be crucial for action recognition, is lost in the process of spatial down-sampling. This raises the first question: Do low-resolution frames contain sufficient information for accurate video recognition?

To answer this question, we quantitatively evaluate the impacts of information loss on recognition accuracy. Specifically, we first obtain frames at different resolutions, $112 \times 112$, $96 \times 96$, $72 \times 72$, $56 \times 56$, by resizing the original $224 \times 224$ frames to simulate information loss in the down-sampling process. Then we up-sample these low-resolution frames back to $224 \times 224$ to exclude the influence brought by scale variations. We train and evaluate video recognition models using these up-sampled

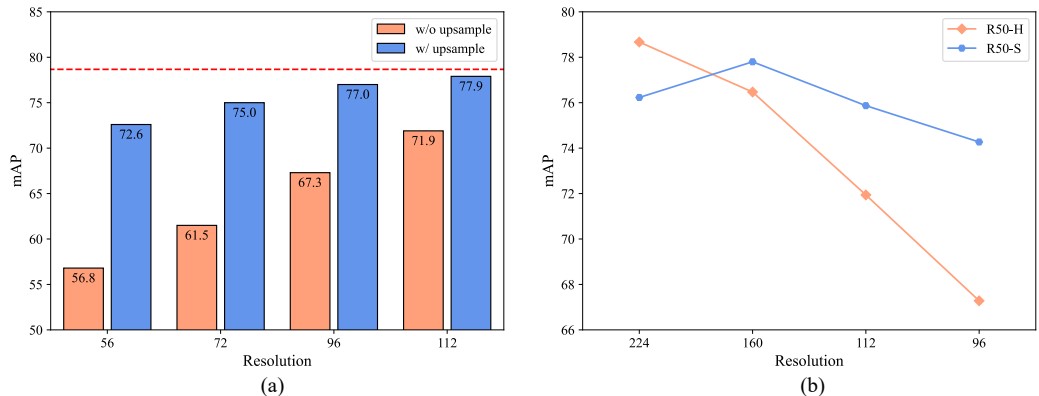

Figure 1: All figures report accuracy of ResNet-50 on ActivityNet. Figure (a) shows the performance of models trained on up-sampled frames and original low-resolution frames, respectively. The dotted red line in figure (a) denotes the performance of model trained on frames of $224 \times 224$ resolution. Figure (b) presents the performances of R50-H and R50-S at different resolutions.

frames. In addition, another group of models which is directly trained and tested on low-resolution frames is included for comparison. The experimental results are shown in Figure 1 (a). In contrast to the common belief, we find that information loss is not the main cause of performance degradation. Even for super low resolutions, e.g., $56 \times 56$, the preserved information can well support highly efficient video recognition with decent performance. This study suggests that high-resolution frames may not be necessary for efficient video recognition, in the sense that they contain highly redundant or task-irrelevant information.

**The mismatch between network and input scale.**   In Figure 1 (a), we notice that input scale is a critical factor affecting performance. But in what way scale variations influence performance is not clear. Inspired by prior work [40] which highlights the importance of coordinating and balancing network architecture with input scales, we hypothesize that networks designed with high-resolution (e.g., $224 \times 224$) inputs in mind may not be good learners on low-resolution frames. That is, the mismatch between network and input scale degrades the performance.

To verify this hypothesis, we adjust the strides of convolution layers at each stage in ResNet-50 from (1, 2, 2, 2) to (1, 1, 2, 2), and compare the performance of the original ResNet-50 (R50-H) with the modified ResNet-50 (R50-S) at different resolutions. As shown in Figure 1 (b), with this simple adjustment, R50-S significantly outperforms R50-H at low resolutions but underperforms R50-H at high resolutions, which is consistent with our speculation.

Based on this observation, a trivial solution to address such performance degradation is to design tailored network architectures for low-resolution video recognition. However, the potential number of resolutions and network variations render this solution costly and inflexible in practice. Is there a general solution for this problem without modifying current networks? Even though networks designed for high-resolution inputs are initially inferior low-resolution learners, we speculate these powerful networks are potentially 'backward compatible' with low-resolution recognition as long as proper guidance is provided. We are thus interested in investigating whether knowledge distillation is sufficient to bridge the performance gap through transferring knowledge or feature representations from a teacher network with high-resolution inputs to a student network with low-resolution inputs.

# 4   Cross-Resolution Knowledge Distillation

In this section, we give an overview of Cross-Resolution Knowledge Distillation (ResKD) and comprehensively study its properties in terms of (i) Effectiveness: How does ResKD compare to other efficient video recognition methods? (ii) Generality: Does ResKD work well for SOTA models with dense sampling? (iii) Scalability: Is ResKD scalable with varying resolutions? (iv) Benefits: How does ResKD help low-resolution video recognition?

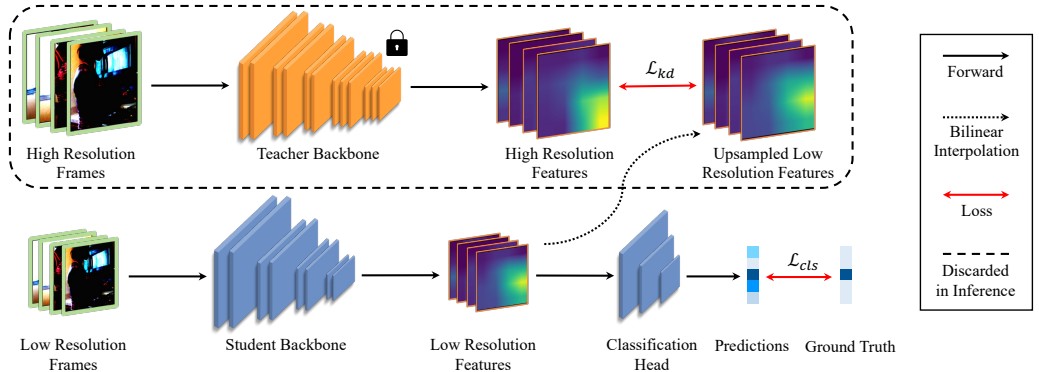

Figure 2: Overview of ResKD: In the training phase, a pre-trained teacher network taking high-resolution frames as input is leveraged to guide the learning of a student network on low-resolution frames. While for evaluation, only the student is deployed to make predictions.

## 4.1 The ResKD Framework

As shown in Figure 3, the ResKD framework exhibits no difference from traditional knowledge distillation [17] except that the teacher and the student take input of different resolutions during training. The idea is quite straightforward: We utilize a teacher good at high-resolution video recognition to guide the learning of the student on low-resolution frames. But instead of utilizing logit KD as in [36, 39], we adopt feature KD to transfer knowledge across resolutions. Although logit KD seems to be a natural choice given the feature maps from the teacher and the student have different spatial sizes, we find it compromises hints in temporal and spatial dimensions which are crucial for cross-resolution knowledge transfer in videos. Please refer to Section 5 for more details. The overall loss function can be written as

$$\mathcal{L} = \mathcal{L}_{cls}\left(\mathbf{y}_s, \mathbf{y}\right) + \alpha \cdot \mathcal{L}_{kd}, \quad \text{where } \mathcal{L}_{kd} = \text{MSE}\left(\mathbf{f}_t, \Phi(\mathbf{f}_s)\right) \tag{1}$$

$\mathcal{L}_{cls}$ is the standard cross-entropy loss between the predictions $\mathbf{y}_s$ of the student and ground-truth labels $\mathbf{y}$. While $\mathbf{f}_t$ and $\mathbf{f}_s$ denote features generated by the teacher and student, respectively. $\Phi$ is a bilinear interpolation operation to align the spatial dimensions of $\mathbf{f}_s$ with $\mathbf{f}_t$. $\text{MSE}(\cdot, \cdot)$ stands for the standard mean squared error.

## 4.2 Benchmark Setups

**Datasets.** We benchmark ResKD on five commonly used action recognition datasets. In particular, ActivityNet-v1.3 [3] and FCVID [18] are used for evaluation on untrimmed videos: (1) ActivityNet-v1.3 contains 10,024 training videos and 4,926 validation videos from 200 action classes, with an average duration of 117 seconds. (2) FCVID includes 45,611 training videos and 45,612 validation videos labeled into 239 classes, with an average length of 167 seconds. As for trimmed video evaluation, we use: (3) Kinetics-400 [19] is a large-scale scene-related dataset covering 400 human action categories, with at least 400 video clips for each class. (4) Mini-Kinetics is a subset of Kinetics-400 introduced by [32, 33]. It includes 121,215 videos for training and 9,867 videos for testing, coming from 200 action classes. (5) Something-Something V2 [14] is a temporal-related dataset which contains 168,913 training videos and 24,777 validation videos over 174 classes.

**Evaluation Protocols.** Our method is evaluated in terms of accuracy and efficiency. For accuracy measurement, we adopt mean average precision (mAP) for multi-label classification on ActivityNet-v1.3 and FCVID, and top-1 accuracy for multi-class classification on remaining datasets. For efficiency measurement, we use Giga floating point operations (GFLOPs) which is hardware-independent to represent computational cost. Since the number of frames used per video varies across different methods and datasets, for fair comparisons, we report GFLOPs per video in the following experiments. Besides, we measure the practical efficiency with throughput on a single Tesla V100 SXM2 GPU.

Table 1: **Comparison with state-of-the-art efficient video recognition methods on ActivityNet-v1.3 and Mini-Kinetics**. MN, MN-T, and RN stand for MobileNetV2, MobileNetV2-TSM, and ResNet, respectively. The best two results are bold-faced and underlined, respectively.

| Methods | Backbones | ActivityNet | | Mini-Kinetics | | FCVID | |
|---|---|---|---|---|---|---|---|
| | | mAP | GFLOPs | Top-1 Acc. | GFLOPs | mAP | GFLOPs |
| LiteEval [49] | MN+RN101 | 72.7% | 95.1 | 61.0% | 99.0 | 80.0% | 94.3 |
| SCSampler [24] | MN+RN50 | 72.9% | 42.0 | 70.8% | 42.0 | 81.0% | 42.0 |
| ListenToLook [11] | MN+RN101 | 72.3% | 81.4 | - | - | - | - |
| AdaFrame [50] | MN+RN101 | 71.5% | 79.0 | - | - | 80.2% | 75.1 |
| AR-Net [32] | MN+RN18,34,50 | 73.8% | 33.5 | 71.7% | 32.0 | 81.3% | 35.1 |
| AdaFuse [33] | RN50 | 73.1% | 61.4 | 72.3% | 23.0 | 81.6% | 45.0 |
| Dynamic-STE [20] | RN18,50 | 75.9% | 30.5 | 72.7% | 18.3 | - | - |
| FrameExit [13] | RN50 | 76.1% | 26.1 | 72.8% | 19.7 | - | - |
| VideoIQ [39] | MN+RN50 | 74.8% | 28.1 | 72.3% | 20.4 | 82.7% | 27.0 |
| AdaFocus [45] | MN+RN50 | 75.0% | 26.6 | 72.2% | 26.6 | 83.4% | 26.6 |
| AdaFocus V2 [46] | RN50 | 78.9% | 34.1 | 74.0% | 34.1 | **84.5%** | 34.1 |
| OCSampler [28] | MN-T+RN50 | 77.2% | 25.8 | 73.7% | 21.6 | 82.7% | 26.8 |
| **ResKD** | RN50 | **80.0%** | **17.4** | **75.4%** | **8.7** | 84.4% | **17.4** |

**Implementation Details.**   Unless otherwise specified, we uniformly sample 8 frames from each trimmed video and 16 frames from each untrimmed video, respectively. For data pre-processing, following [27, 45, 28], we first apply random scaling to all sampled frames, then augment them with $224 \times 224$ random cropping and random flipping in the training stage. For the input to student, we further down-sample the resolution of video frames to $112 \times 112$. During inference, we resize the short side of all frames to 128 while keeping the aspect ratio, then center-crop them to $112 \times 112$. By default, we adopt ResNet-152 [15] as the teacher network, ResNet-50 [15] as the student network, and $112 \times 112$ as the student input resolution. If not mentioned, we perform feature KD on the last-layer outputs of the teacher and the student. We use the codebase provided by [5] for implementation.

### 4.3   How does ResKD compare to other efficient video recognition methods?

We start by comparing the efficiency of ResKD with state-of-the-art efficient video recognition methods. As presented in Table 1, ResKD substantially surpasses all alternative methods by achieving superior accuracy with $1.5\times$-$5.5\times$ computation savings on ActivityNet and FCVID. In particular, ResKD outperforms AR-Net which also tries to increase efficiency with low resolution frames, by 6.2% mAP on ActivityNet, while using half of the GFLOPs.

It is worth noting that ResKD demonstrates even higher efficiency on trimmed videos. On Mini-Kinetics, ResKD requires only 40% of computation of the recently proposed method OCSampler, while excelling OCSampler by 1.7% top-1 accuracy. Remarkable success on short-range videos demonstrates the robustness of ResKD against temporal redundancy variance in videos, which gives the probability of combing ResKD with existing temporal-redundancy-removing methods, e.g., OCSampler, to achieve higher efficiency.

Additionally, we benchmark ResKD against state-of-the-art efficient CNNs proposed for video recognition. TSM-ResNet50 is adopted as the backbone of teacher and student network on Something Something V2. As shown in Table 2, without using a cumbersome teacher, ResKD-TSM uses 73% less computation (8.7 v.s. 32.7 GFLOPs) but achieves 1.4% higher top-1 accuracy than baseline TSM. The results verify the effectiveness of ResKD and further confirm the great potential of low-resolution frames in efficient video recognition.

### 4.4   Does ResKD work well for SOTA models with dense sampling?

Although numerous efficient video recognition methods have been proposed, most of them are limited to efficient video recognition settings, i.e., using light-weight 2D-CNNs as backbones and sparsely sampled frames as input. We believe it will make more sense if ResKD could serve as a general acceleration method for state-of-the-art video recognition. Therefore, we take one step forward to evaluate the effectiveness of ResKD on 3D-CNNs and transformers, with densely sampled frames.

Table 2: **Comparison with SOTA efficient CNNs on Something Something V2.** MN and R50 stand for MobileNetV2 and ResNet-50. The best two results are bold-faced and underlined, respectively.

| Methods | Backbones | Top-1 Acc. | GFLOPs |
|---------|-----------|------------|--------|
| TSN[44] | R50 | 27.8% | 33.2 |
| TRN$_{\text{RGB/Flow}}$ [52] | BN-Inc. | 55.5% | 32.0 |
| TANet[31] | R50 | 60.5% | 33.0 |
| TEA[25] | R50 | **60.9%** | 35.0 |
| TSM[27] | R50 | 59.1% | 32.7 |
| AdaFuse-TSM[33] | R50 | 59.8% | 31.3 |
| AdaFocus-TSM[45] | MN+R50 | 59.7% | 23.5 |
| AdaFocusV2-TSM[46] | MN+R50 | 59.6% | 18.5 |
| **ResKD-TSM** | R50 | 60.6% | **8.7** |

SlowOnly [10] and Video Swin [29] are chosen as the representatives of 3D-CNN and transformer for evaluation, respectively. Following practice in [10, 29], we densely sample $8 \times 10 \times 3$ frames for test on SlowOnly and $32 \times 4 \times 3$ frames for test on Video Swin. The results are reported on Kinetics-400 in Table 3. One can observe that ResKD largely closes the performance gap between high-resolution and low-resolution input on both architectures. Especially, significant savings in computation (up to 1218 less GFLOPs per video) is achieved with little sacrifice in accuracy.

Table 3: **Effectiveness of ResKD on SlowOnly and Video Swin.** ResKD-SlowOnly uses SlowOnly-ResNet50 as the teacher and student networks. ResKD-Swin_S uses Swin_B as the teacher and Swin_S as the student. Since Swin_B and Swin_S have different numbers of output channels, some adjustments are made to ResKD, which is discussed in detail in Appendix. Numbers in brackets denote the input resolution.

| Models | Top-1 Acc. | GFLOPs |
|--------|------------|--------|
| SlowOnly (224) | 74.5% | 1260 |
| SlowOnly (112) | 69.3% | 332 |
| ResKD-SlowOnly | 73.1% | 332 |

| Models | Top-1 Acc. | GFLOPs |
|--------|------------|--------|
| Swin_S (224) | 80.1% | 1639 |
| Swin_S (112) | 76.3% | 421 |
| ResKD-Swin_S | 80.0% | 421 |

## 4.5 Is ResKD scalable with varying resolutions?

In this subsection, we access the scalability of ResKD in terms of accuracy and efficiency by changing the input resolutions.

**Scalability in terms of accuracy.** We change input resolution within {56x56, 72x72, 96x96, 112x112, 144x144}, and plot the corresponding mAP v.s. GFLOPs trade-off curves on ActivityNet in Figure 3. In comparison with state-of-the-art methods, ResKD consistently achieves better trade-off at various computational costs. Besides, different from other methods whose accuracy drops drastically with low computational budgets, ResKD maintains decent performance even at very low resolutions.

**Scalability in terms of practical efficiency.** As pointed out in [28], the design of many efficient methods cannot fully leverage the parallel computing power of current devices. Consequently, fewer GFLOPs do not necessarily imply faster inference speed. To verify that the practical efficiency of ResKD is also scalable with input resolution, we measure the inference speed of ResKD with different input resolutions under the same hardware setup. It can be seen from Table 4 that reductions in GFLOPs bring continuous and stable boosts in inference speed.

## 4.6 How does ResKD help low-resolution video recognition?

In the following, we analyze the benefits of ResKD from a qualitative perspective, with a special focus on how it alleviates the mismatch between network architecture and input scale.

Table 4: **Scalability of inference speed regarding input resolutions.** Throughput (number of videos processed per second) is measured on a single Tesla V100 SXM2 GPU with the batch size of 64.

| Resolution | 224p | 144p | 112p | 96p | 72p | 56p |
|---|---|---|---|---|---|---|
| mAP | 81.7% | 81.3% | 80.0% | 76.2% | 73.4% | 70.5% |
| GFLOPs | 65.9 | 28.3 (2.3↓) | 17.4 (3.8↓) | 12.1 (5.4↓) | 8.3 (7.9↓) | 4.8 (13.7↓) |
| Throughput | 78.7 | 163.2 (2.1↑) | 262.8 (3.3↑) | 333.9 (4.2↑) | 434.7 (5.5↑) | 631.5 (8.0↑) |

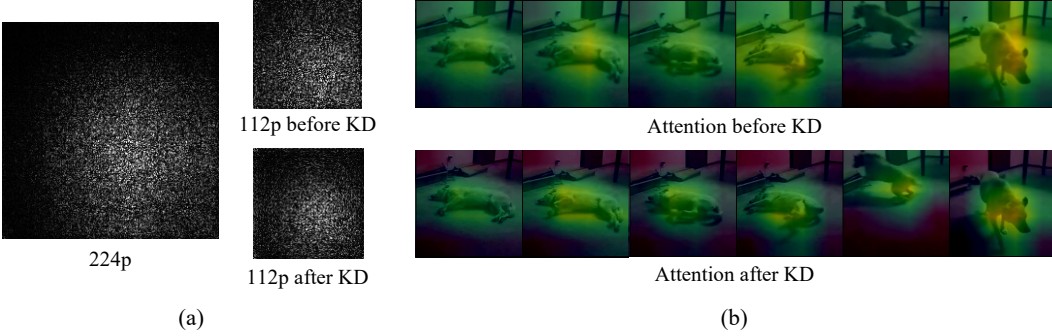

|       |       |
|-------|-------|
| 112p before KD |  |
| 224p | 112p after KD |
| (a)  | Attention before KD / Attention after KD / (b) |

Figure 4: **Visualization before and after ResKD.** In figure (a), we choose a pixel close to the bottom right from the last-layer feature map for ERF visualization, as there is no central pixel in the feature map of 112p input. In figure (b), the focused region is highlighted in bright yellow.

**Shrinking effective receptive field.** As shown in Section 3, the mismatch between network and input size is the main cause of performance degradation for low-resolution recognition. It is also observed that decreasing the down-sampling ratio of network or up-scaling the intermediate feature maps can largely alleviate this problem. Essentially, these operations shrink the amplified receptive fields of the network on low-resolution inputs. We therefore hypothesize ResKD has similar effects on the student network. To verify this hypothesis, we compare the effective receptive fields (ERF) of the student network before and after ResKD. In Figure 4 (a), one can observe that the ERF contracts significantly after ResKD, resembling the original ERF on high-resolution inputs.

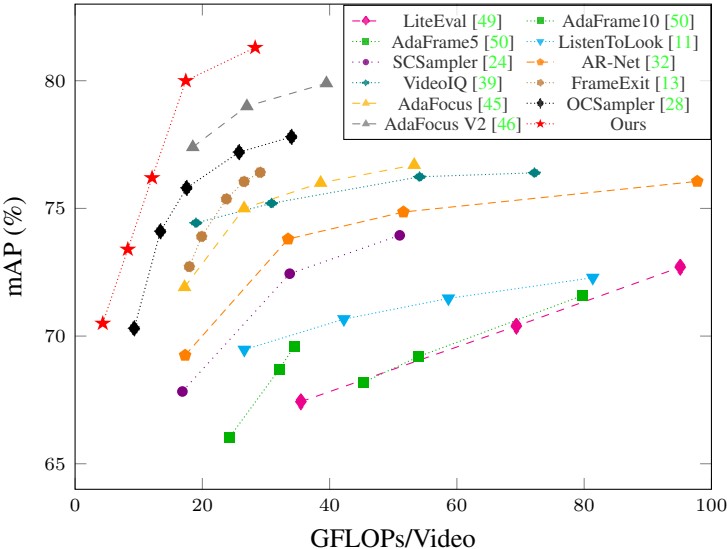

Figure 3: **Accuracy v.s. efficiency trade-off curves.**

**Improving localization.** Another interesting finding is that the model after ResKD demonstrated stronger localization capability on low-resolution inputs. Following practice in [22], we visualize the activation-based attention of the student. From the demos in Figure 4 (b), it can be seen that model with ResKD can better focus its attention on objects or actions of interest, while the attention of model without ResKD spreads across larger areas including uninformative backgrounds. Based on these observations, we conjecture that ResKD helps low-resolution video recognition by contracting the ERF to match object scales in low-resolution inputs.

Table 5: **Ablation on design choices of ResKD.** KL Div. and MSE stand for Kullback–Leibler Divergence and Mean Squared Error, respectively. We report the results on ActivityNet, with ResNet-50 as student backbone and ResNet-152 as teacher backbone.

| KD method | Supervision signal | Loss type | Temporal dim. | Spatial dim. | mAP |
|---|---|---|---|---|---|
| Baseline | N/A | N/A | ✗ | ✗ | 71.8% |
| Clip-level KD | cls. score | KL Div. | ✗ | ✗ | 73.4% |
| Frame-level KD | cls. score | KL Div. | ✓ | ✗ | 76.6% |
| Pixel-level KD | feature map | MSE | ✓ | ✓ | 78.5% |

Table 6: **Ablation on model distillation and resolution distillation.** R152 and R50 stand for ResNet-152 and ResNet-50, respectively.

| Settings | Teacher | High resolution | mAP |
|---|---|---|---|
| Baseline | N/A | ✗ | 71.8% |
| Model distill. | R152 | ✗ | 73.3% |
| Resolution distill. | R50 | ✓ | 76.2% |
| ResKD | R152 | ✓ | 78.5% |

## 5 Ablation Study

**Design choices of ResKD**   We investigate how the specific design of KD impacts the performance of ResKD. Depending on the type of supervision signals from the teacher, we categorize KD into (i) Clip-level KD which uses classification score of the whole clip as supervision, (ii) Frame-level KD which uses frame-wise classification score as supervision, (iii) and Pixel-level KD which is supervised on features before global average pooling. Detailed information on these design choices is presented in Table 5. Notably, traditional KD, i.e., Clip-level KD, only brings mild gains in cross-resolution KD settings. But adding temporal dimension supervision, i.e., Frame-level KD, significantly boosts the effectiveness of KD by more than 3% in mAP. A possible explanation for this result is that high frequency supervision provides more hints because frames are noisy in nature. In addition, the results of Pixel-level KD indicate that supervision in spatial dimensions is also crucial for cross-resolution knowledge transfer.

**Model distillation *v.s.* Resolution distillation**   To fully exert the potential of knowledge distillation, we employ a teacher model with a higher capacity than the student. One may argue that performance gains brought by ResKD is mainly attributed to knowledge transfer from a high-capacity teacher network rather than high-resolution input. To show that knowledge from high-resolution frames is the main factor contributing to the efficiency of ResKD, we conduct an ablation on two settings, namely model distillation, and resolution distillation. In model distillation, we use a high-capacity network (ResNet-152) as the teacher but feed both teacher and student with low-resolution ($112 \times 112$) frames during training. In resolution distillation, the teacher network is the same as the student (ResNet-50) but takes high-resolution ($224 \times 224$) frames as input. Ablation results in Table 6 verify the effectiveness of resolution distillation.

## 6 Conclusion

In this paper, we systematically study how to make the most of low-resolution frames in the context of efficient video recognition. We identify that information in high-resolution frames is actually redundant, which can be alleviated by simple spatial down-sampling. However, the mismatch between network architecture and input scale prevents us from fully exploiting the potential of low-resolution frames. To address this issue, we leverage cross-resolution knowledge distillation (ResKD) to guide the learning of the student on low-resolution frames with a teacher expert in high-resolution recognition. Extensive experiments demonstrate that ResKD can serve as an effective, general, and scalable inference acceleration method for state-of-the-art video recognition.

**Acknowledgments.** This work has been supported by Hong Kong Research Grant Council - Early Career Scheme (Grant No. 27209621), HKU Startup Fund, and HKU Seed Fund for Basic Research. Ping Luo is supported by the General Research Fund of HK No.27208720, No.17212120, and No.17200622.

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
