

Figure 5: Figure (a) demonstrates the supervision signals of traditional feature KD and logit KD. Figure (b) illustrates the proposed pixel-level logit KD for general ResKD. $H$ and $W$ refer to the spatial dimensions of features, while $C$ and $K$ refer to the channel number and class number, respectively. GAP stands for global average pooling.

## A Appendix

### A.1 A More General Form of ResKD

By default, ResKD applies feature KD to transfer knowledge across resolutions. In case of the teacher and the student may have different number of output channels, we provide an alternative KD method, namely pixel-level logit KD to replace feature KD. As illustrated in Figure 5, pixel-level logit KD skips the common pooling operation preceding the classification head to obtain pixel-level classification scores, which are then used for knowledge distillation. Such simple adjustments enable it to combine the advantages of feature KD and logit KD. On the one hand, it preserves spatial information for ResKD, which is shown to be beneficial in Section 5 of the main paper. On the other hand, it aligns the channel of features between the teacher and the student, without introducing additional learnable modules to transform the features [37, 21]. The effectiveness of pixel-level logit KD is verified with experiments on Video Swin in Section 4.4 of the main paper.

### A.2 Implementation Details

**Training Hyper-parameters for ResNet** In our implementation, we first pre-train the teacher network using an SGD optimizer with cosine learning rate annealing and a Nesterov momentum of 0.9 for 50 epochs [15, 27, 32, 28, 45]. The size of the mini-batch is set to 64, while the weight decay is set to 1e-4. The training settings of the student are basically the same as the teacher, except that we train the student for 100 epochs on FCVID, Mini-Kinetics, Something Something V2, and 200 epochs on ActivityNet. Both the teacher (ResNet-152 or ResNet-50) and the student (ResNet-50) are initialized with ImageNet pre-trained parameters from the official models provided by PyTorch. The weight of knowledge distillation loss is set to $\alpha = 100$. Notably, during knowledge distillation, we unfreeze the batch normalization layer in the teacher network as suggested in [16].

In ablation studies and experiments in Section 3, we use the same set of aforementioned hyper-parameters except that we only train the student for 50 epochs.

**Training Hyper-parameters for SlowOnly and Video Swin** We follow the configurations in [5] to train SlowOnly-teacher and SlowOnly-student. The weight of knowledge distillation loss is set to $\alpha = 100$. For experiments on Video Swin, we use pre-trained Swin_B provided in the official implementation for Video Swin Transformers as the teacher. We train the student Swin_S initialized with ImageNet pre-trained parameters for 60 epochs, using default configurations in [29]. Pixel-level logit KD is applied for ResKD between Swin_B and Swin_S, where the weight of KD loss is set to $\alpha = 500$ and the temperature is set to $T = 4$.

### A.3 Limitations of ResKD

As a spatial-redundancy-removing method, ResKD lacks particular treatment for temporal redundancy of input, making it incapable of fully addressing redundancy issues such as temporal redundancy on untrimmed videos. Sparse sampling strategy [44] is employed in most of our experiments to alleviate

this issue. While another potential solution is to combine ResKD with other temporal-redundancy-removing methods. For example, one may leverage selected frames from adaptive frame selection methods as input to ResKD, so that ResKD can further compress resolution of frames to achieve higher efficiency. In fact, how to combine ResKD with existing efficient video recognition methods can be an interesting topic, e.g., could ResKD be applied to the skim network of current adaptive methods so that the skim network could provide higher-quality features to the policy network in identifying important frames? We leave this for future exploration.