# OpenReview forum: "Rethinking Resolution in the Context of Efficient Video Recognition"
_NeurIPS.cc/2022/Conference — NeurIPS 2022 Accept_

### Official Review · Reviewer_THu4 · 2022-06-29

**Rating:** 6
**Confidence:** 3
**Soundness:** 3 good
**Presentation:** 3 good
**Contribution:** 2 fair

**Summary:**

This paper proposes resolution distillation for efficient video recognition (ResKD).
ResKD adopts the typical knowledge distillation framework, which consists of a teacher network (with high resolution) and a student network (with low resolution).
ResKD calculates the MSE loss between teacher network feature and the upsampled student network feature for knowledge distillation.
The authors conducted extensive experiments to validate the efficacy of this framework.

**Questions:**

1. Ablation study: In this paper, you present the results of using the last-layer feature for ResKD, have you tried features at other layers or to use features at multiple stages for ResKD?
2. What's the recognition performance for this setting: Dataset: Kinetics400, Teacher: Swin_S (224), Student: Swin_S (112).

**Limitations:**

Yes.

**Strengths And Weaknesses:**

## Strengths
1. ResKD is a simple & effective idea for efficient video understanding.
2. Extensive experiments have been conducted to validate the effectiveness of this framework.

## Weakness
1. This paper simply adopts the typical knowledge distillation framework, which lacks novelty to some extent.
2. The section 3 of this paper seems to be trivial and not closely related to ResKD.
   1. If the backbone is unchanged and the spatial resolution is reduced, the recognition performance may severely suffer from the over early down-sampling and the drastically reduced computation amounts (4x reduced for 224 -> 112, *etc.*).
   2. However, if you remove a down-sampling layer in the early stage, the computation amounts would be close to the original network with high resolution. Thus that's not a good idea for efficient video understanding.

In summary, I think this paper is a good empirical study, but lacks novelty.

---

> ### Author Response · Authors · 2022-08-02
> **Thanks for your comments**
>
> Dear Reviewer THu4,
>
> We really appreciate your comments and your support for our work. We hope our response can address your concerns.
>
> 1. "The section 3 of this paper seems to be trivial and not closely related to ResKD."
> - **Clarification of the controlled experiments in Section 3**. It seems your concern regarding the trivial aspect of Section 3 is mostly related to the experimental designs and results in Figure 1, so we make the following clarification on the insights of these experiments:
>   - The first experiment (Figure 1.(a)) is not merely to say spatially down-sampling frame resoluton will cause drastic performance degradation. But more importantly, we reveal that information loss, e.g., loss of high-frequency signals, during the down-sampling process is not the main cause of degradation. This conclusion is drawn from the observation that re-up-sampling low-resolution frames can close most of the gap with original high-resolution input.
>   - In the second experiment (Figure 1.(b)), R50-S (the one with a removed early down-sampling layer) outperforms R50-H on low-resolution inputs. However, the advantage of R50-S over R50-H is not simply attributed to more computations in R50-S, as R50-H outperforms R50-S at high resolutions (e.g., 224p) with less computation. Instead, the result suggests that the architecture of R50-S is more compatible with low-resolution input. Considering most existing networks are designed with high-resolution input in mind, we argue that mismatch between network architecture and input size can be an important factor of underperformance on low-resolution frames.
> - **The relation between Section 3 and ResKD**. Section 3 provides the underlying motivation for applying ResKD, which consists of two key ideas:
>   - Low-resolution frames (e.g., 112p) still contain sufficient information for accurate recognition, which serves as the basis for ResKD. (The quality of low-resolution frames poses an upper-bound for ResKD. Note that the rationale of ResKD is to guide the learning of the student on low-resolution input. But if low-resolution frames do not contain sufficient visual clues, there is not much ResKD can do.)
>   - Mismatch between network architecture and input size is a main cause of performance degradation. This is the direct motivation of ResKD, which is proposed to minimize the gap between network and input size. Section 4.6 provides more explanations on how ResKD helps close the gap.
>
> 2. Lack novelty as the paper simply adopts the typical knowledge distillation framework.
> - The design of ResKD is simple but not trivial. We have identified and overcome some major flaws in previous methods (e.g., [19], [34]) applying cross-resolution KD on videos, which brings up to 5.2% gains in mAP on ActivityNet v2 (in Table 5 of ablation study).
> - As an empirical study, our work does not aim to be exactly a “new-method paper”. In contrast, we try to keep the design of framework as simple as possible to make the idea clear -- compressing spatial resolution of videos can be a practical and powerful way to boost video recognition efficiency.
> - As the name suggests, the contribution of this paper largely lies in the empirical findings, which have not been presented in literature of efficient video recognition:
>   - Resolution is an important source of redundancy. Low-resolution frames have great potential for efficient video recognition.
>   - The main cause of performance degradation on low-resolution frames is not information loss in the down-sampling process, but the mismatch between network architecture and input size.
>   - Cross-resolution KD can serve as a strong baseline to fulfill the potential of low-resolution frames.
>   - Keeping spatial and temporal hints from the teacher is critical for the success of cross-resolution KD in videos.
>
> 3. **Ablation on intermediate layer KD.**
> This is quite an interesting question. Actually, we have experimented with both KD at early stages and multi-layer KD. However, we empirically found adding features from early stages for knowledge distillation brings no good or even slightly decreases the performance. The detailed results can be found in the table below.
> Number in the "Stages" row means the output features from the corresponding stage of ResNet-50/152 are used for ResKD. The results are reported on ActivityNet v2. We set the weight of KD loss to 100 for all stages.
>
>
> >| Stages | &nbsp;&nbsp; --   |  [1]  |  [2]  |  [3]  |  [4]  | [1, 2, 3, 4] |
> >|:------:|:-----:|:-----:|:-----:|:-----:|:-----:|:------------:|
> >|  mAP   | 71.8% | 71.0% | 71.5% | 71.7% | 78.5% |    78.2%     |
>
> 4. **Results of  Swin_S (224) -> Swin_S (112).**
> Without using a large-scale teacher (Swin_B), the gain from cross-resolution KD is still considerable.
>
> >|   &nbsp;&nbsp;Teacher    |   Student    | mAP   |
> >|:------------:|:------------:| ----- |
> >|      --      | Swin_S (112) | 76.3% |
> >| **Swin_S (224)** | **Swin_S (112)** | **78.8%** |
> >| Swin_B (224) | Swin_S (112) | 80.0% |

---

> ### Author Response · Authors · 2022-08-09
> **Welcoming further discussions**
>
> Dear Reviewer THu4,
>
> Thank you for your feedback. As we are approaching the end of the discussion period, we would like to ask whether there are any remaining concerns regarding our paper or our response? We are happy to answer any further questions.
>
> We sincerely thank you for your efforts in reviewing our paper and your suggestions for strengthening the experimental part. We will add the ablation study mentioned above into the latter version of the paper. If you find our responses have addressed all of your concerns, would you like to kindly raise the rating?

---

### Official Review · Reviewer_AzGj · 2022-07-05

**Rating:** 6
**Confidence:** 4
**Soundness:** 3 good
**Presentation:** 4 excellent
**Contribution:** 3 good

**Summary:**

This paper analyzed the underlying causes of performance degradation on low-resolution frames, and proposed a cross-resolution knowledge distillation (ResKD) to bridge the gap between network and input size. This work is interesting.

**Questions:**

In Section 4.5, it is not clear how the student Backbone is trained for each different low resolution.

**Limitations:**

NA.

**Strengths And Weaknesses:**

The problems are clearly stated, and the manuscript is well written.
Some experimental conditions are not clearly described.

---

> ### Author Response · Authors · 2022-08-02
> **Thanks for your appreciation to our work**
>
> Dear Reviewer AzGj,
>
> We really appreciate your comments for our work and your support for acceptance. We hope our response can address your concerns.
>
> **The training settings in Section 4.5:** The training of the student backbone in Section 4.5 for each different low resolution is almost the same as the default settings stated in Section 4.2 (e.g., the same teacher and hyper-parameters). The only difference is the resolution of input of the student.

---

> > ### Comment · Reviewer_AzGj · 2022-08-09
> > **Thanks for the authors' reply**
> >
> > The authors have addressed my concern. I have no further question.

---

### Official Review · Reviewer_uP8D · 2022-07-11

**Rating:** 4
**Confidence:** 5
**Soundness:** 3 good
**Presentation:** 3 good
**Contribution:** 2 fair

**Summary:**

This paper proposes to improve efficient video recognition on low-resolution frames by cross-resolution knowledge distillation (ResKD). Specifically, the student network with shallower architecture and lower-resolution frames is guided by a teacher network with deeper architecture and higher-resolution inputs. By distilling the learnt knowledge from teacher network, ResKD narrows the performance gap between efficient model and large model. The experiments are conducted on several public video recognition datasets. The results validate the good performance-efficiency trade-off achieved by ResKD.

**Questions:**

My main concerns are novelty and insights. Please see the details in Strengths and Weaknesses part.

**Limitations:**

The limitations and potential negative societal impact are well discussed in the paper.

**Strengths And Weaknesses:**

Strengths:

+ The paper is mostly clear and easy to follow.
+ The good performance on several public datasets validate the effectiveness of ResKD.

Weaknesses:

- Novelty. The basic idea of this paper is to distill the knowledge from large-scale teacher model. However, utilizing the popular knowledge distillation (KD) for video recognition is not new (e.g., [34], [35]). It is not surprising that KD from a powerful model could benefit efficient models. Especially, the KD between networks with different input frame resolutions for video recognition has been proposed in [34]. This paper is more like a technical report that validates the existing KD technique on efficient video recognition. Given these points, the technical contribution of this paper is limited.
- Insights on video data. Considering this paper mainly focuses on video recognition, some insights on video data should be given. However, the proposed ResKD ignores the temporal redundancy in videos and only studies the spatial redundancy. ResKD is more like a general low-resolution image classification framework and has no special design for video data.
- Table 1. In Table 1, the depth of ResNet (in Backbones column) should be listed. The performance of ResKD is lower than AdaFocus V2 on FCVID, however, ResKD is marked as the best result, which may mislead the readers.
- Throughput comparison. This paper only lists the throughput in Table 4. Considering that GFLOPs cannot fairly measure the efficiency of a network, the throughput comparisons between ResKD and baseline methods should be given.

---

> ### Author Response · Authors · 2022-08-02
> **Response (2/2)**
>
> 4. "This paper is more like a technical report that validates the existing KD technique on efficient video recognition."
>
> - We hope responses 1&2&3 can address most of the concerns regarding novelty of the paper.
> - Besides, we emphasize that, as an exploratory work, the goal of this paper is to reveal the great potential of low-resolution frames in efficient settings and provide a strong but simple baseline to fulfill the potential. Looking into current literature [42, 26, 43, 47, 30, 36], there seems to be a common belief that low-resolution frames alone cannot support efficient video recognition. For instance, low-resolution frames are mostly used to provide side information of a video (e.g., help identify salient frames), while accurate recognition still highly relies on high-resolution frames, which consumes most of the computational budget. In this sense, our work challenges this common belief and encourages future works to rethink the usage of low-resolution frames.
>
> 5. Lack insights on video data
> - Firstly, we point out that **spatial redundancy itself is an important topic in efficient video recognition (e.g., [42], [43])**. The performance gains further confirm the significance of our study on spatial resolution in videos. In recent years, huge progress has been made to alleviate temporal redundancy in videos. For example, OCSampler [26] is able to compress a short video of thousands of frames into 6 frames, without sacrificing much accuracy. However, further compressing the temporal dimension will risk a much higher chance of missing key frames, leading to rapidly deteriorating trade-off between efficiency and accuracy. For this reason, we turn to exploring spatial redundancy of video data to further boost efficiency.
> - We do not propose further complexities in the framework to deal with temporal redundancy, as it would interfere with the message we try to convey - compressing spatial resolution of videos can be a practical and powerful way to boost video recognition efficiency.
> - Moreover, our developed ResKD closely correlates with video data and demonstrates temporal dimension information should be maintained during KD. One distinction between early work (e.g., [34]) applying cross-resolution KD on videos and ResKD is that **KD in ResKD is frame-based**. As shown in Table 5, switching from clip-level KD to frame-level KD brings more than 3% increment in mAP on ActivityNet v2. A possible explanation to this result is that high-frequency temporal supervision provides more hints because frames are noisy in nature. This is an important new finding that we believe is instrumental to future research.
> - Although ResKD only explores spatial redundancy, potentially it can be combined with existing methods working on temporal redundancy to explore benefits of both sides (e.g., use OCSampler [26] to sample frames and ResKD to compress spatial resolution of sampled frames).
>
> 6. **Table 1.**
> Thank you for making these suggestions to help us increase clarity of Table 1. We have revised Table 1 following your advice in the new version of the paper. BTW, the reason we originally highlighted ResKD as the best result on FCVID is in consideration of both accuracy and efficiency (ResKD achieves very similar performance, but only uses half of the computation as AdaFocus V2).
>
> 7. **Throughput comparison.**
> Thank you for helping us strengthen the experiments. As many baseline methods are not open-source, we only find AdaFocus and AdaFocus V2 for comparison. We report results on ActivityNet v2. Throughput (number of videos processed per second) is measured on a single Tesla V100 SXM2 GPU with the batch size of 64.
>
>
> >|   &nbsp;&nbsp;&nbsp; Method    |  mAP  | GFLOPs | Throughput (videos/s) |
> >|:-----------:|:-----:|:------:|:---------------------:|
> >|  AdaFocus   | 75.0% |  26.6  |          72           |
> >| AdaFocus V2 | 78.9% |  34.1  |          100          |
> >|    ResKD    | 80.0% |  17.4  |          263          |
> >
> >  Notably, ResKD demonstrates even stronger efficiency when evaluated with throughput than GFLOPs. This is because most adaptive methods like AdaFocus are not fully parallel in computation, i.e., they formulate the frame/patch selection policy as a sequential decision task as pointed out in [26]. While ResKD does not suffer from such problems for its simple design.

---

> > ### Comment · Reviewer_uP8D · 2022-08-08
> > **Thanks for your reply**
> >
> > I would like to thank the authors for their rebuttal and additional evaluations, as well as further discussion of the novelty. The rebuttal addressed part of my concerns. However, the performance of ResNetKD without deep teacher network (76.2%) is lower than AdaFocus V2 and OCSampler as shown in Table 1. It is hard to say “it still achieves SOTA performance.” Moreover, the rebuttal says “our work does not aim to be exactly a new-method paper”, which further confuses me about the novelty of this paper. Utilizing the typical KD framework but changing the teacher/student networks lacks enough novelty as a NeurIPS paper to me. Given these points, I maintain my original rating of Borderline Reject.

---

> > > ### Author Response · Authors · 2022-08-08
> > > **Further clarification on the performance and novelty**
> > >
> > > Dear Reviewer uP8D,
> > >
> > > Thanks for pointing out your confusion. We want to clarify some misunderstandings that caused some of your concerns.
> > >
> > > 1. Performance without deep teacher network
> > > - We emphasize that, as we are studying efficient video recognition, **both accuracy and efficiency should be taken into consideration** to evaluate a method. You mentioned AdaFocus V2 achieved higher accuracy. However, this is achieved at the cost of efficiency (e.g., the number of FLOPs required by AdaFocus V2 doubles compared with ResKD). Actually, from Figure 3 in this paper, it can be shown that **at the same computational budget (17.4 GFLOPs)**, 76.2% is a SOTA result (e.g., well above OCSampler [26]).
> > > - Due to the time limit of rebuttal period, we only trained the student network for 50 epochs. The training time is much shorter than our default settings, i.e., 200 epochs. From our experience, longer training will further benefit the student, e.g., the accuracy of student R50 increases by 1.5% when increasing the training epochs from 50 to 200, with R152 as a teacher.
> > >
> > > 2. Novelty
> > > - This paper does not aim to be exactly a new-method paper because **the goal of this paper is to reveal the great potential of low-resolution frames in efficient settings and provide a strong but simple baseline to fulfill the potential**. We believe that is where the value of empirical study papers lies, i.e., challenging common beliefs and identifying surprising results of the method.
> > > - **From the empirical perspective**, this paper investigates the underlying causes of performance degradation on low-resolution frames, revealing their great potential in efficient settings. **From the technical perspective**, the proposed cross-**Res**olution **KD** (ResKD) overcomes some major flaws of previous KD methods to significantly boost the effectiveness of cross-resolution KD for the first time. For these reasons, we feel it is a bit unfair to simply summarize the contribution of this paper as "utilizing the typical KD framework but changing the teacher/student networks".

---

> ### Author Response · Authors · 2022-08-02
> **Response (1/2)**
>
> Dear Reviewer uP8D,
>
> We really appreciate your comments. We hope our point-to-point response can address your concerns and clarify our contribution.
>
> 1. "The basic idea of this paper is to distill the knowledge from large-scale teacher model."
>
> - Firstly, we clarify that the basic idea of ResKD is **not to distill knowledge from a large-scale teacher model but from high-resolution frames**. We have discussed in detail the motivations for adopting cross-resolution KD in Section 3. Moreover, ablation studies in Table 6 validate that cross-resolution KD contributes the most to the performance gains. As shown in the table below, without using large-scale networks as the teacher, ResKD still achieves SOTA performance on ActivityNet v2 (better mAP and fewer GFLOPs).
> - The key point of this paper is not limited to studying cross-resolution KD in efficient video recognition. **More importantly**, we provide a comprehensive and in-depth review of resolution and efficiency in the context of video recognition. Our empirical study does reveal some **counter-intuitive findings**, e.g., the main cause of performance degradation on low-resolution frames is not information loss in the down-sampling process, but mismatch between network architecture and input size. Based on this observation, we propose ResKD. The strong performance of ResKD not only validates the effectiveness of cross-resolution KD in efficient video recognition, but also **demonstrates the great potential** of low-resolution frames in trade-off for efficiency, which is largely overlooked by current literature.
>
> >|  &nbsp;Method  | Teacher Backbone | Backbone  |  mAP  | GFLOPs |
> >|:--------:|:----------------:|:---------:|:-----:|:------:|
> >| AdaFocus |        NA        | ResNet-50 | 75.0% |  26.6  |
> >|  ResKD   |    ResNet-50     | ResNet-50 | 76.2% |  17.4  |
>
>
>
> 2. "It is not surprising that KD from a powerful model could benefit efficient models."
>
> - As discussed in response #1, a "powerful" teacher is not a pre-requisite of the high performance.
> - Our work is the first to show cross-resolution KD can **massively boost** the performance of efficient video recognition. Previous methods [19, 34] simply adopt logit KD for cross-resolution KD on videos, which seems to be a natural choice given the feature maps from the teacher and the student have different spatial sizes. However, we point out in Section 5, **traditional logit KD compromises hints in temporal and spatial dimensions** which are crucial for cross-resolution knowledge transfer in videos. As a result, early work[19] only achieved **minor improvements** with KD (**0.2% mAP** on ActivityNet v2, in Table 5 of [19]). Similar cases can be observed in [34].
> - In contrast, we propose pixel-level feature KD. Albeit simple, it performs surprisingly well as shown in ablation studies in Table 5 (switching from clip-level logit KD to pixel-level feature KD results in **5.2% mAP** gain on ActivityNet v2). The large performance gain and simplicity of the design reveal that keeping spatial and temporal hints from the teacher is one key for cross-resolution KD in videos. Otherwise, only minor improvements can be obtained by vanilla KD.
>
> >|   &nbsp;&nbsp;&nbsp;&nbsp;&nbsp;&nbsp;&nbsp;Method       | KD design  | Temporal hint | Spatial hint |
> >|:------------------:|:----------:|:-------------:|:------------:|
> >| Didik, et al. [34] |  Logit KD  |       N       |      N       |
> >|  Dynamic-STE [19]  |  Logit KD  |       Y       |      N       |
> >|       ResKD        | Feature KD |       Y       |      Y       |
>
>
> 3. There exists work applying cross-resolution KD in video recognition.
>
> - As discussed in response #1, the focus and contribution of this paper are beyond studying cross-resolution KD in video recognition. Instead, we are talking about **a more general question** -- Is down-sampling spatial resolution of videos a promising solution to boosting efficiency? Our in-depth analysis on this question reveals some novel findings challenging the common belief. Besides, we use extensive experiments and strong results to give an "yes" answer to the question, which **has not been addressed in early works**.
> - As discussed in response #2, the design of ResKD is different from previous works applying cross-resolution KD, which makes a fundamental difference in performance. From this perspective, we think the fact "there exists work applying cross-resolution KD in video recognition" does not undermine the contribution of this paper. Instead, it proves the technical contribution of this paper is non-trivial as no existing works conduct in-depth studies on effective strategies for cross-resolution KD in videos. The ablation studies on KD design, analysis, and high performance further strengthen our contributions.

---

> ### Author Response · Authors · 2022-08-08
> **Need More Clarification?**
>
> Dear Reviewer uP8D,
>
> Thank you for your feedback. We feel like we have addressed all of your questions from the initial reviews in our response. As we are approaching the end of the discussion period, we would like to ask whether there are any remaining concerns regarding our paper or our response? We are happy to answer any further questions.

---

### Meta-Review · Area_Chair_8y5L · 2022-08-27

**Recommendation:** Accept
**Confidence:** Certain

**Metareview:**

After the rebuttal and discussion, two reviewers recommend acceptance, one borderline rejection. Most concerns of the raised in the borderline review were addressed at a sufficient detail in the rebuttal. The AC sees no reason the reject this paper.

**Award:**

No

---

### Decision · Program_Chairs · 2022-09-14

Accept